# Spatial distribution of complete basic childhood vaccination and associated factors among children aged 12–23 months in Ethiopia. A spatial and multilevel analysis

**Getanew Aschalew Tesfa**[1]*, **Delelegn Emwodew Yehualashet**[1], **Addisu Getnet**[2], **Kirubel Biweta Bimer**[3], **Binyam Tariku Seboka**[1]

1 School of Public Health, College of Health Science and Medicine, Dilla University, Dilla, Ethiopia,
2 Department of Midwifery, College of Health Science and Medicine, Dilla University, Dilla, Ethiopia,
3 Department of Nursing, College of Health Science and Medicine, Dilla University, Dilla, Ethiopia

* getanewaschalew@gmail.com

**Data Availability Statement:** Data is available online from https://www.dhsprogram.com/data. A letter of permission for the use of the data was

## Abstract

### Background

Complete childhood vaccination considerably aids in the reduction of morbidity and mortality from vaccine-preventable childhood diseases. Understanding the geographical disparity of complete basic childhood vaccination and Identifying associated factors is vital to designing appropriate interventions. This study aimed to assess the spatial distribution and associated factors of complete basic childhood vaccination among children aged 12–23 months in Ethiopia.

### Methods

A two-stage stratified sampling technique was used based on the 2019 Ethiopian mini demographic and health survey data. A total weighted sample of 1,028 children was included in the analysis. ArcGIS version 10.8 software was used to visualize the spatial distribution of complete basic childhood vaccination. The Bernoulli-based model was used to detect significant clusters of areas using SaTScan version 9.6 software. To identify associated factors, multilevel logistic regression analyses were used, and all variables with a p-value less than 0.05 were reported as statistically significant predictors.

### Results

Complete basic childhood vaccination among children aged 12–23 months had a significant variation in Ethiopia (Moran's I = 0.276, p<0.001). The spatial scan analysis identified the most likely significant primary clusters with low complete basic childhood vaccination coverage in the Somali region's Afder, Liben, Shabelle, and Nogobe zones; the Southern Nation Nationality and Peoples Region's (SNNPR) Gedeo and Sidama zones; and the Oromia region's Bale and Guji zones. The second significant cluster was found in the Afar region's zones 1, 4, and 5, as well as the northern Somali region's Siti zone. In the multivariable

secured from DHS program. We used 2019 EMDHS dataset. The data is publicly available and anyone can access it through reasonable request. The sole requirements for accessing the Demographic and Health Survey (DHS) program datasets are registration and a justification of the request's objective.

**Funding:** The Authors received no specific funding for this work.

**Competing interests:** The authors have declared that no competing interests exist.

**Abbreviations:** ANC, Antenatal care; AOR, Adjusted odds ratio; BCG, Bacillus Calmette Guerin; CI, Confidence interval; CSA, Central statistical agency; DHS, Demographic and health survey; DPT, Diphtheria, Tetanus, and Pertussis; EA, Enumeration area; EDHS, Ethiopian demographic and health survey; EMDHS, Ethiopian mini demographic and health survey; EPI, Expanded Program on Immunization; LLR, Log-likelihood Ratio; OR, Odds Ratio; RR, relative risk; WHO, World Health Organization.

multilevel analysis, maternal age 20–24, 25–29, 35–39, and 40–44 years, delivery at a health facility, four or more antenatal care (ANC) visits, orthodox religion fellowship, maternal primary education, Muslim religion fellowship, living in the Afar, Somalia, and Oromia regions, and living in rural areas were all found to be significantly associated with complete basic childhood vaccination.

## Conclusion

A geographically significant variation of complete basic childhood vaccination was observed. Maternal age, maternal education, religion, place of delivery, ANC visit, region, and residence were significantly associated with complete basic childhood vaccination. Developing immunization campaigns targeting areas that had low basic vaccination coverage and designing healthcare programs that can motivate facility-based delivery and ANC follow-up is recommended.

## Introduction

Immunization is one of the most cost-effective and impactful available interventions in public health [1]. Complete vaccination of infants and children has a significant benefit in the prevention of morbidity and mortality from childhood disease [2, 3]. Currently, immunization prevents 2–3 million deaths every year from diseases such as diphtheria, pertussis, tetanus, influenza, and measles, and 86% of infants received three doses of the DPT3 (diphtheria, tetanus, pertussis) vaccine worldwide, in 2019 [4]. World Health Organization (WHO) and UNICEF (United Nations Children's Fund) warn of a decline in vaccination during covid-19, in 2019, nearly 14 million children missed out on life-saving vaccines such as DPT3, and measles; Two-third of them are in low and middle-income countries namely: Ethiopia, Pakistan, Philippines, Nigeria, Mexico, India, Indonesia, Brazil, the Democratic Republic of the Congo, and Brazil [4].

Global immunization coverage has increased during the past decades [5]. The African region still fallen lags behind other regions of the world in access to vaccines and approximately one in five children don't receive all the necessary and basic vaccines [6]. As a result, over half a million children die from vaccine-preventable diseases annually [7]. Basic childhood immunization coverage in Sub-Saharan African countries is low (59.4%), with disparities between countries [8]. In east Africa, complete basic childhood vaccination is low (69.21% in 2016) ranging from 39.5% in Ethiopia to 85% in Burundi, and remains a major public health issue [9].

In Ethiopia, the major health problem of the country remains preventable communicable diseases including vaccine-preventable diseases for children and nutritional disorders [10]. The Ethiopian Expanded Program on Immunization (EPI) was first launched in 1980, to reduce vaccine-preventable maternal and child morbidity and mortality [11]. An integrated effort was made by the government and the expanded program on immunization for the prevention and control of communicable diseases [10].

Childhood vaccination in Ethiopia is being given on a routine and outreach basis and the immunization schedule strictly follows the WHO recommendations for developing countries; infants should be vaccinated with one dose of BCG at birth or as soon as possible, while the three doses of pentavalent (DPT-HepB-Hib) and polio vaccines (at approximately age 6,10, and 14 weeks), and measles (MCV1) vaccine at the age of 9 months [12, 13]. The EPI

in Ethiopia considers a child to have received all basic vaccinations if he/she has also received three doses of the pneumococcal conjugate vaccine (PCV) at age 6,10, and 14 weeks and two doses of the rotavirus vaccine (at age 6 and 10 weeks) [14]. According to the Ethiopian demographic and health survey (EDHS) report, there is an improvement in complete basic childhood vaccination representing one dose of BCG, three doses of pentavalent (DPT-HepB-Hib), three doses of polio, and one dose of measles vaccine among children aged 12–23 months in the country, increased from 24% in 2011, 39% in 2016, to 44% in 2019 [14–16]. However, complete basic childhood vaccination status is yet very low compared with the national (at least 90% nationally with 80% in each district) and WHO ($> =$ 90%) targets [17].

Previous studies conducted in Ethiopia focused on the prevalence and associated factors of childhood vaccination at the subnational level, with limited studies at the national level. Understanding the geographical disparity of complete basic childhood vaccination using a spatial analysis has become essential to developing focused public health interventions. Therefore, this study aimed to investigate the spatial distribution and associated factors of complete basic childhood vaccination among children aged 12–23 months in Ethiopia using national representative data. The result of this study might be helpful for policymakers and program managers in augmenting complete basic childhood vaccination coverage.

## Methods and materials

### Study design, setting, and data source

This study was conducted using secondary data, the Ethiopian mini demographic and health survey 2019 (EMDHS), retrieved from https://www.dhsprogram.com/data. EMDHS 2019 was a population-based cross-sectional study conducted from March 21, 2019, to June 28, 2019, using a representative sample across the country and it is the second mini-demographic and health survey [15]. Ethiopia is located in the horn of Africa (3˚–14˚ N and 33˚–48˚ E). Administratively, it has nine regional states (Tigray, Afar, Amhara, Oromia, Somalia, Benishangul-Gumuz, Southern Nation Nationalities, and Peoples Region (SNNPR), Gambela and Harari) and two city administrations (Addis Ababa and Dire Dawa). Each region is further subdivided into zones, districts, towns, and kebeles (the lowest unit) [18].

### Sample size and sampling procedure

The sampling frame used for the EMDHS 2019 is a frame of all census enumeration areas created for the Ethiopian population and housing census (PHC) which was conducted by the central statistical agency (CSA) in 2019. During the census, each kebele was subdivided into enumeration areas (EAs). The 2019 EMDHS used a two-stage stratified cluster sampling. In the first stage, a total of 305 EAs (212 in rural areas and 93 in urban areas) were selected with probability proportional to EA size based on the 2019 PHC frame and with independent selection in each sampling stratum. A household listing operation was carried out in all of the selected EAs and served as a sampling frame for the selection of households in the second stage. In the second stage, a fixed number of 30 households per cluster were selected with an equal probability of systematic selection from the newly created household listing [15]. In this study, a total weighted sample of 1028 living children aged 12–23 months with their mothers was included.

### Variables of the study

**Outcome variable.** Complete basic childhood vaccination status of children aged 12–23 months was the outcome variable of the study, which has a binary outcome coded as "yes" if

the child received one BCG vaccine, three doses of the polio vaccine, three doses of pentavalent vaccine, and one dose of measles vaccine and "no" if the child failed to take the recommended doses of vaccine. Information on vaccination status was obtained in three ways: from written vaccination cards, the mothers' verbal reports, and health facility records.

**Independent variables.** The predictor variables of the study were categorized into individual-level factors including sociodemographic variables such as maternal age (categorized as 15–19, 20–24, 25–29, 30–34, 35–39, 40–45), marital status (categorized as never married, married, and widowed/divorced/separated), sex of household head (categorized as male and female), sex of the child (categorized as male and female), maternal education (no formal education, primary, and secondary and above), wealth index (poor, middle, and rich), and Child and maternal related characteristics such as place of delivery (categorized as home delivery and health facility), mode of delivery(cesarean delivery and vaginal delivery), birth order (categorized as 1$^{st}$, 2$^{nd}$ -5$^{th}$, and > = 6), ANC visit (no ANC visit, 1–3 visit, and > = 4 visits).

Community-level factors such as residence (categorized as urban and rural), and region (categorized as Addis Ababa, Afar, Amhara, Oromia, Beneshangul, Gambela, SNNPR, Somali, Harari, Tigray, and Dire Dawa).

## Data management and analysis

After we accessed the data; data extraction, cleaning, weighting, and recoding were done before any statistical analysis. Descriptive statistics were done to describe the characteristics of the study population using STATA. ArcGIS version 10.7 and Spatial Scan Statistics (SaTScan TM version 9.6) software were used to execute the spatial data analysis. To determine the presence of spatial autocorrelation, Global Moran's Index (Moran's I) was used. Moran's I is a statistic that measures whether complete basic childhood vaccination patterns were clustered, dispersed, or randomly distributed in the study area by producing an output ranging from -1 to +1. Moran's I value close to -1 indicated complete basic childhood vaccination dispersed, whereas Moran's I +1 indicated complete basic childhood vaccination clustered, and complete basic childhood vaccination distributed randomly if Moran's I value was zero. A statistically significant Moran's I (P-value < 0.05) indicated the presence of spatial autocorrelation and can lead to the rejection of the null hypothesis (complete basic childhood vaccination is randomly distributed). Spatial scan analyses were applied to identify the most likely significant clusters. In the SaTScan, a Bernoulli-based model was used and the scanning window with maximum likelihood was the most likely significant cluster, a Log-likelihood ratio (LLR), and p-value were assigned to these clusters.

To rule out a substantial association between variables, a multi-collinearity test was done using the variance inflation factor (VIF). There is no multi-collinearity because all variables have VIF less than 5 and tolerance greater than 0.1. The intra-cluster correlation coefficient (ICC) was used to assess cluster variation. To identify factors associated with complete basic childhood vaccination, we used a multilevel logistic regression model to take into account the hierarchical nature of the DHS data. After fitting a bi-variable multilevel logistic regression analysis, variables with a p-value of < 0.2 in the bi-variable analysis were further considered in the multivariable multilevel logistic regression analysis. For the multilevel logistic regression, four models were constructed. The null model was the first model with no independent variables to detect the extent of the possible contextual effect; model I (a model with only individual-level factors), model II (which was adjusted for community-level factors), and model III (containing both individual and community level factors). Adjusted odds ratio (AOR) with 95% CI and p-value<0.05 in the multivariable multilevel logistic regression model were used to declare statistically significant variables associated with complete basic childhood vaccination.

### Ethical issues

This study was based on secondary data analysis, and permission to download and use the data for our research was obtained from the measure DHS program. As a result, ethical approval and participant consent are not necessary for this particular study. From preliminary, the data were collected by "The DHS program" by taking informed consent from individuals. The dataset is publicly available anonymously in the DHS program's official database.

## Results

### Sociodemographic and socioeconomic characteristics of the study participants

A total weighted sample of 1028 children aged 12–23 months was included in the analysis. 399 (38.8%) of the total children were between the ages of 12 and 15 months and 533 (51.8%) of them were female. The majority of the children, 359(34.9%) were born to mothers aged 25–29. Regarding region, four hundred five (39.4%), two hundred eighteen (21.2%), and one hundred ninety-nine (19.4%) were from Oromia, Amhara, and Southern Nation Nationalities and Peoples Region (SNNPR) respectively. Of the total, 715 (69.5%) lived in rural areas and 383 (37.3%) of the child's mothers were Orthodox Christian religion followers. 431 (41.9%) were from poor household wealth status (Table 1).

### Maternal and child health-related characteristics

Among the total participants, 439 (44.0%) were born to mothers who had greater than or equal to four ANC visits, and nearly half of them, 551(53.6%) were born in the health facility. Regarding birth order, the majority of them, 598 (58.2) were in the second to fifth birth order. Concerning the mode of delivery, 70 (6.8%) were born by cesarean delivery (Table 2).

### Basic childhood vaccination coverage

The overall complete basic childhood vaccination coverage among children aged 12–23 months in Ethiopia was 44%. The percentage of children who had all of their basic childhood vaccinations varied by area, ranging from 83% in Addis Ababa to 73% in Tigray, one-fifth (20%) in Afar, and 19% in the Somalia region. In terms of specific vaccine utilization at the national level; nearly three-fourths (72.96%) of the children received BCG, 60.9% pentavalent third dose, 59.94% polio third dose, and 58.53% measles first dose vaccine (Fig 1).

### Spatial analysis result of complete basic childhood vaccination

Spatial distribution of complete basic childhood vaccination: A low proportion of complete basic childhood vaccination was observed almost in the entire Somali region, Afar, Northwest Gambella, western and eastern parts of SNNPR, some southwest parts of Amhara, and Oromia regions ranging from 0%-20%. However, high coverage of complete basic childhood vaccination was observed in Addis Ababa, central and western Tigray, northeast Amhara, some parts of SNNP, and the Oromia regions of Ethiopia (Fig 2).

The global spatial autocorrelation analysis based on attribute values and feature locations revealed the spatial distribution of complete basic childhood vaccination among children aged 12–23 months in Ethiopia was non-random (Global Moran's I = 0.275696, p-value = 0.000). The result has shown that the observed Moran's Index value (0.275696) was greater than the expected Index (-0.003509) and the p-value is <0.05, which is statistically significant. Given

**Table 1. Sociodemographic and socioeconomics characteristics of the study participants.**

| Variables | Weighted frequency | Percent |
|---|---|---|
| Region | | |
| Tigray | 77 | 7.53 |
| Afar | 15 | 1.46 |
| Amhara | 218 | 21.19 |
| Oromia | 405 | 39.41 |
| Somali | 56 | 5.43 |
| Benishanguel | 11 | 1.04 |
| SNNP | 199 | 19.38 |
| Gambela | 4 | 0.40 |
| Harari | 3 | 0.25 |
| Addis Ababa | 34 | 3.32 |
| Dire Dawa | 6 | 0.59 |
| Residence | | |
| Urban | 313 | 30.47 |
| Rural | 715 | 69.53 |
| Maternal age | | |
| 15–19 | 73 | 7.06 |
| 20–24 | 237 | 23.03 |
| 25–29 | 359 | 34.89 |
| 30–34 | 160 | 15.56 |
| 35–39 | 134 | 13.04 |
| 40–44 | 53 | 5.16 |
| 45–49 | 13 | 1.26 |
| Maternal educational status | | |
| No formal education | 464 | 45.14 |
| Primary | 418 | 40.66 |
| Secondary and above | 146 | 14.20 |
| Marital status | | |
| Never in union | 2 | 0.2 |
| married | 983 | 95.62 |
| divorced/widowed/separated | 43 | 4.18 |
| sex of household head | | |
| male | 887 | 86.28 |
| female | 141 | 13.72 |
| Religion | | |
| Orthodox | 383 | 37.28 |
| Protestant | 265 | 25.80 |
| Muslim | 350 | 34.09 |
| Traditional and other | 29 | 2.84 |
| Household wealth index | | |
| Poor | 431 | 41.89 |
| Middle | 179 | 17.37 |
| Rich | 419 | 40.74 |
| sex of child | | |
| Male | 495 | 48.17 |
| Female | 533 | 51.83 |
| Child age in months | | |
| 12–15 | 399 | 38.84 |
| 16–19 | 334 | 32.45 |
| 20–23 | 295 | 28.71 |

**Table 2. Maternal and child health-related characteristics.**

| Variables | Weighted frequency | Weighted % |
|---|---|---|
| ANC visit | | |
| No visit | 251 | 25.22 |
| 1–3 | 306 | 30.74 |
| > = 4 visits | 439 | 44.04 |
| Mode of delivery | | |
| Caesarian delivery | 70 | 6.83 |
| Vaginal delivery | 958 | 93.17 |
| Birth order | | |
| 1st | 242 | 23.53 |
| 2–5 | 598 | 58.17 |
| > = 6 | 188 | 18.31 |
| Place of delivery | | |
| Home | 477 | 46.36 |
| Health facility | 551 | 53.64 |

the Z-score of 5.330904 Indicates there is a less than 1% likelihood that this clustered pattern could be the result of random chance (Fig 3).

**Spatial scan statistical analysis result.** The spatial scan analysis found 61 significant clusters of areas with low complete basic childhood vaccination rates, indicating that complete basic childhood immunization rates are lower inside the SaTScan window than outside the SaTScan window. Of these, 24 were primary clusters. The primary clusters were located in Somali (Afder, Liben, Shabelle, and Nogobe zones), SNNPR (Gedeo and Sidama zones), and Oromia region (Bale, and Guji zones) centered at 4.028421 N, 41.180723 E with 409.92 km radius, a relative risk (RR) of 1.65, and LLR of 23.58 at p = 0.001. The second most likely significant clusters were located in almost the entire Afar (zone 1, zone 4, and zone 5) and northern Somalia (Siti zone) regions centered at 11.558430 N, 41.440210 E) with 172.92 km radius, RR of 1.69, and LLR of 18.45 at p = 0.001.

The third most significant cluster was located in southern Afar (zone 3) and Oromia (North Shewa and west Harerge zones) region centered at 9.073631 N, 40.134473 E with 118.15 km radius, RR of 1.63, and LLR of 10.90 at p = 0.001. While, the fourth significant cluster with low complete basic childhood vaccination was detected in the northern Somalia

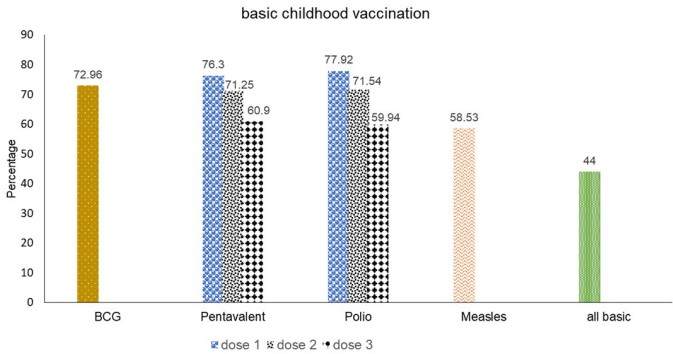

**Fig 1. Prevalence of complete basic childhood vaccination and its components among children aged 12–23 months in Ethiopia, EMDHS 2019.**

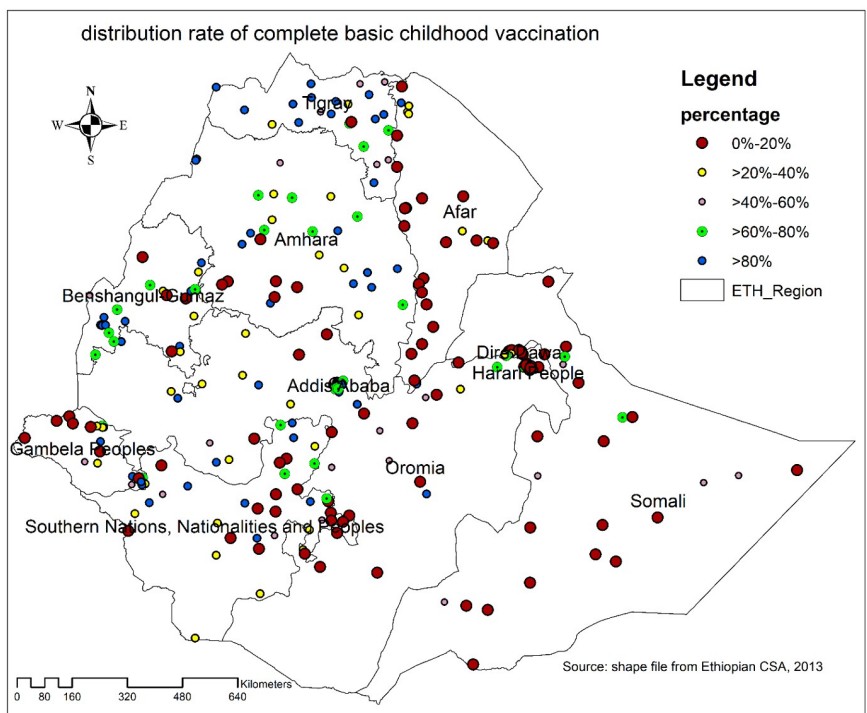

**Fig 2. Spatial distribution of complete basic childhood vaccination among children aged 12–23 months across regions of Ethiopia, EMDHS 2019.**

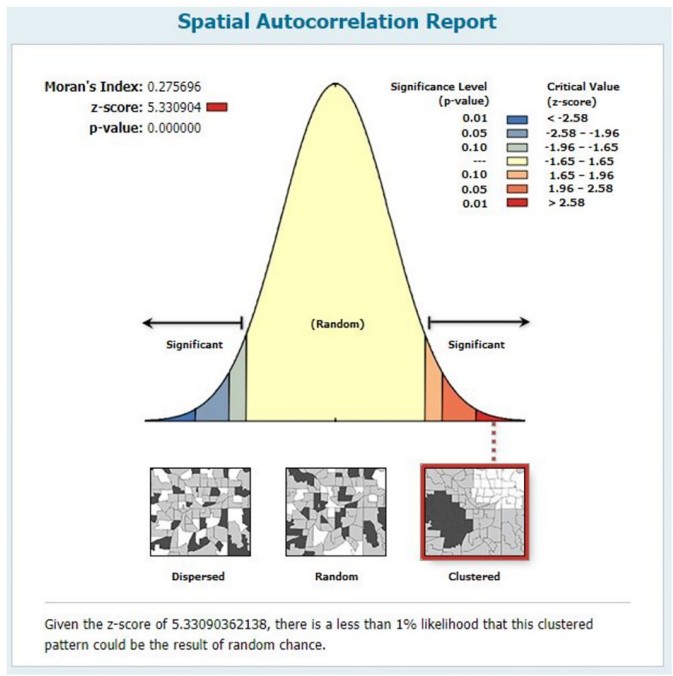

**Fig 3. Spatial autocorrelation report of complete basic childhood vaccination among children aged 12–23 months in Ethiopia, EMDHS 2019.**

**Table 3. SaTScan analysis of complete basic childhood vaccination among children aged 12–23 months in Ethiopia.**

| Cluster type | Significant EAs (clusters) detected | Coordinates/radius | Population | cases | RR | LLR | p-value |
|---|---|---|---|---|---|---|---|
| 1 | 144, 141, 125, 143, 142, 114, 136, 138, 137, 111, 89, 113, 123, 110, 183, 117, 134, 188, 186, 172, 181 | (4.028421 N, 41.180723 E)/ 409.92 km | 95 | 83 | 1.65 | 23.58 | 0.001 |
| 2 | 26, 32, 31, 30, 33, 34, 126, 47, 45, 48, 49, 44, 29, 46, 50 | (11.558430 N, 41.440210 E)/ 172.92 km | 58 | 53 | 1.69 | 18.48 | |
| 3 | 41, 28, 105, 88, 42, 40, 102, 106, 127, 43, 90 | (9.073631 N, 40.134473 E)/ 118.15 km | 39 | 35 | 1.63 | 10.9 | 0.001 |
| 4 | 128, 130, 121, 109, 107, 129, 254, 255, 249, 251, 248 | (9.673818 N, 42.836549 E)/84.49 km | 44 | 38 | 1.57 | 9.67 | 0.006 |
| 5 | 220, 218, 229, 230, 219 | (8.356129 N, 33.766046 E)/70.96 km | 26 | 24 | 1.67 | 8.59 | 0.04 |

Notes: RR = relative risk, LLR = log likelihood ratio.

region (Fafan zone) and eastern Oromia region (east Harerge zone) centered at 9.673818 N, 42.836549 E) with 84.49 km radius, RR of 1.57, and LLR of 9.67 at p = 0.006 and the fifth clustered located in Nuer and Kelem Welega zones centered at 8.356129 N, 33.766046 E with 70.96 km, RR of 1.67, and LLR = 8.59 at p = 0.04 (Table 3 and Fig 4).

## Factors associated with complete basic childhood vaccination

Multivariable multilevel logistic regression analyses were fitted to identify the significant individual and community-level predictor variables associated with complete basic childhood vaccination. In the full model, which included all individual and community level variables, the age of the mother, place of delivery, ANC visits, region, religion, and place of residency were factors significantly associated with complete basic childhood vaccinations among children aged 12–23 months.

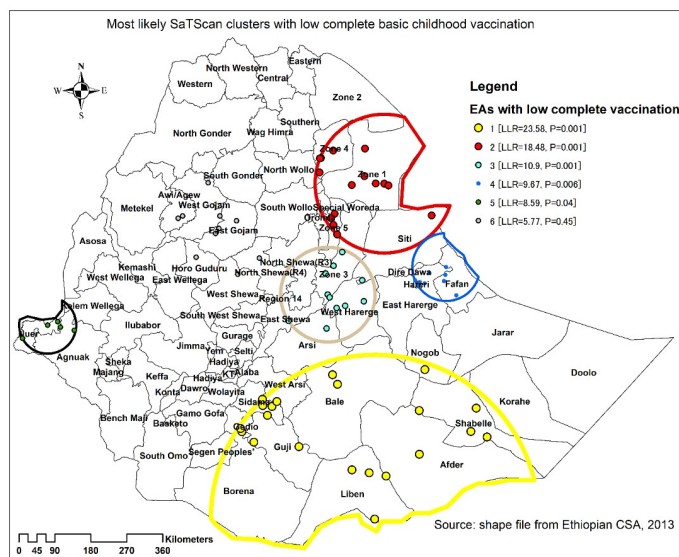

**Fig 4. Spatial scan analysis of complete basic childhood vaccination among children aged 12–23 months in Ethiopia.**

**Individual-level factors.**   Maternal age was significantly associated with complete basic childhood vaccination. Children born to mothers who were 20–24 years old were approximately two times (AOR = 2.12, 9S%CI: 1.06–4.24) more likely to be fully vaccinated than children whose mothers were 15–19 years old. Children born from mothers who attained primary education were 1.67 times more likely (AOR = 1.67, 95% CI: 1.13–2.47) to be fully vaccinated than children whose mothers didn't attain any formal education.

The odds of being completely vaccinated among children whose mothers were orthodox Christian and Muslim religion followers were 2.17 times (AOR = 2.17, 95%CI: 1.23–3.84) and 2.25 times (AOR = 2.25, 95%CI: 1.30–3.89) higher log-likelihood of fully vaccinated than whose mothers were protestant religion followers respectively. Children born in the health facility were 1.85 times (AOR = 1.85, 95%CI: 1.27–2.71) higher odds of being completely vaccinated than children born at home. Children born to mothers who had more than or equal to four ANC visits were approximately two times more likely to have complete basic childhood vaccination than children born to women who did not have ANC visits (AOR = 2.2, 95% CI: 1.03–4.73) (Table 4).

**Community-level factors.**   The odds of complete basic childhood vaccination among children living in the Afar region were 87% times (AOR = 0.13, 95%CI: 0.05–0.34), Somalia region 79% times (AOR = 0.21, 95%CI: 0.07–0.58), and Oromia region 60% times (AOR = 0.40, 95% CI: 0.16–0.99) less likely to be vaccinated compared with children living in Addis Ababa respectively. Children residing in rural areas were 49% times (AOR = 0.51, 95%CI: 0.32–0.80) less likely to be fully vaccinated than children residing in urban areas (Table 4).

**Multilevel analysis (random effect analysis).**   The result shows that the complete basic childhood vaccination rate was not similarly distributed across the communities. According to the ICC value, cluster/EA variability accounted for 47.3% of overall complete basic childhood vaccination variability in the null model. In the full model, 18.4% of the chances of complete basic childhood vaccination variation were detected across the communities. The MOR for complete basic childhood vaccination was 5.15 in the null model, showing that there was variation between communities. The unexplained community variation in complete basic childhood vaccination decreased to 2.3 in the final model. And still, the effect of clustering is statistically significant in the full model when all factors are included. Deviance was used to verify model comparability and fitness. The fourth model (the full model) was chosen as the final best-fitted model since it has the minimum deviation value (Table 4).

## Discussion

Complete basic childhood vaccination in Ethiopia among children aged 12–23 months was low (44%). This result was in line with the study done in the Wonago district [19], and Mozambique [20]. But, lower than the study done in east Africa (69.2%) [9], and a study in northwest Ethiopia [21]. This might be due to that this study was done based on national-level data and thus a high variability of immunization services among regions is the possible reason for the observed differences [22]. Additionally, health system differences and access to information on maternal and child health might be a possible explanation [23]. Complete basic childhood vaccination among children aged 12 to 23 months varied significantly between Ethiopian regions, according to this study. It was low in Afar, Somalia, eastern SNNPR, and northeast Oromia areas. While, high complete basic childhood vaccination was found in Addis Ababa, western Tigray, central Tigray, northern Oromia, central Oromia, and western Benishanguel-Gumuz regions.

In the multivariable multilevel analysis maternal age, region, maternal education, place of delivery, ANC visits, religion, and residence were significantly associated with complete basic

**Table 4. Multivariable multilevel analysis of complete basic childhood vaccination among children aged 12–23 months.**

| Variables | Null model | Model I (AOR 95%CI) | Model II (AOR 95%CI) | Model III (AOR 95%CI) |
|---|---|---|---|---|
| Maternal age | | | | |
| 15–19 | | 1 | | 1 |
| 20–24 | | 1.84(0.94–3.62) | | 2.12(1.06–4.24)* |
| 25–29 | | 2.46(1.27–4.75) | - - - - | 2.78(1.42–5.47)** |
| 30–34 | | 1.86(0.92–3.79) | | 1.78(0.86–3.68) |
| 35–39 | | 3.83(1.79–8.20) | | 4.34(1.98–9.48)** |
| 40–44 | | 3.63(1.43–9.24) | | 3.65(1.40–9.47)** |
| 45–49 | | 1.45(0.37–5.66) | | 1.62(0.43–6.16) |
| Maternal education | | | | |
| No education | | 1 | - - - - - | 1 |
| Primary | | 1.74(1.20–2.52) | | 1.67(1.13–2.47)** |
| Secondary and above | | 1.47(0.89–2.42) | | 1.41(0.83–2.39) |
| Religion | | | | |
| Protestant | | 1 | | 1 |
| Orthodox | | 3.63(2.34–5.63) | - - - - | 2.17(1.23–3.84)** |
| Muslim | | 1.63(1.07–2.48) | | 2.25(1.30–3.89)** |
| Traditional and other | | 0.81(0.27–2.43) | | 0.84(0.28–2.51) |
| Household wealth index | | | | |
| Poor | | 1 | | 1 |
| Middle | | 0.81(0.51–1.28) | - - - - | 0.75(0.47–1.20) |
| Rich | | 1.43(0.98–2.06) | | 1.06(0.69–1.65) |
| ANC visit | | | | |
| No visit | | 1 | | 1 |
| 1–3 | | 1.67(0.81–3.43) | - - - - | 1.47(0.70–3.10) |
| > = 4 | | 2.80(1.35–5.83) | | 2.2(1.03–4.73)* |
| Mode of delivery | | | | |
| Caesarian delivery | | 1 | - - - - | 1 |
| Vaginal delivery | | 0.82(0.45–1.47) | | 0.79(0.43–1.45) |
| Place of delivery | | | | |
| Home | | 1 | | 1 |
| Health facility | | 2.23(1.56–3.20) | - - - | 1.85(1.27–2.71)** |
| Region | | | | |
| Addis Ababa | | | 1 | 1 |
| Afar | | | 0.09(0.04–0.21) | 0.13(0.05–0.34)** |
| Amhara | | | 0.72(0.31–1.65) | 0.95(0.38–2.36) |
| Oromia | - - - - | - - - - | 0.23(0.10–0.51) | 0.40(0.16–0.99) * |
| Somali | | | 0.10(0.40–0.25) | 0.21(0.07–0.58)** |
| Benishangule | | | 0.88(0.38–2.08) | 1.03(0.41–2.60) |
| SNNPR | | | 0.33(0.14–0.74) | 0.68(0.27–1.72) |
| Gambela | | | 0.26(0.11–0.62) | 0.52(0.20–1.37) |
| Harari | | | 0.35(0.15–0.81) | 0.43(0.17–1.06) |
| Tigray | | | 1.32(0.56–3.09) | 1.56(0.61–3.98) |
| Dire Dawa | | | 0.50(0.22–1.14) | 0.62(0.25–1.53) |
| Residence | | | | |
| Urban | - - - - | - - - - - | 1 | 1 |
| Rural | | | 0.31(0.22–0.45) | 0.51(0.32–0.80)** |
| Model fitness and comparison | | | | |

*(Continued)*

**Table 4.** (Continued)

| Variables | Null model | Model I (AOR 95%CI) | Model II (AOR 95%CI) | Model III (AOR 95%CI) |
|---|---|---|---|---|
| Variance | 2.950871 | 1.078944 | 1.15517 | .7395723 |
| ICC (%) | 47.3 | 24.7 | 26.0 | 18.4 |
| PCV (%) | Ref. | 63.44 | 60.85 | 74.94 |
| MOR | 5.15 | 2.7 | 2.8 | 2.3 |
| LLR | -630.71424 | -528.02119 | -568.73318 | -507.60068 |
| Deviance | 1,261.42848 | 1,056.04238 | 1,137.46636 | 1,015.20136 |

Notes:

* = p-value< = 0.05,

** = p-value< = 0.01,

1 = reference group, AOR = adjusted odds ratio, null model = without the predictors, model I = adjusted for individual factors, model II = adjusted for community-level factors, model III = adjusted for both individual level and community level factors.

childhood vaccination. Mothers aged 20–24, mothers aged 25–29, mothers aged 35–39, and mothers aged 40–44 were more likely to vaccinate their children compared with mothers aged 15–19. The finding of this study was related to the study done in east Africa [9], and a study done in Afghanistan [24]. This might be due to that younger parents may have relied on the approval or decision of their older family members. in contrast, older mothers may have a sense of responsibility and accumulated knowledge about immunization services in the modern healthcare system [25].

This study showed that maternal education was a significant predictor of complete basic childhood vaccination, children born to mothers who attained education had higher odds of being fully vaccinated than children whose mothers didn't attain any formal education. Supported by a systematic review done in 45 countries [26], a study conducted in Uganda [27], and a study done in Sekota district, northern Ethiopia [28]. This could be because when a woman's educational standing increases, so does her ability to make decisions, as well as increasing access to health information, changing attitudes, and addressing some of the negative cultural behaviors that prevent children from receiving basic vaccines [27]. In contrast to this result, the study done in the Oromia region has shown complete basic childhood vaccination didn't have any statistically significant association with maternal educational level [29]. The variation in the study setting and sample size could be one explanation for the discrepancy. This study was based on a nationally representative sample, whereas the prior study in the Oromia region exclusively included children from the Ambo district. Children whose mothers had ANC follow up were more likely to be completely vaccinated than those whose mothers didn't have ANC follow up. This result was supported by the study done in sub-Saharan African countries [29], and a study done in Ethiopia [30]. Awareness created by healthcare professionals about basic immunization services when they meet mothers during follow-up could be the possible explanation for this [31].

There is a statistically significant relationship between complete basic childhood vaccination among children living in urban and rural areas. Children living in urban areas had higher odds of being fully vaccinated than children living in rural areas. Supported by a study done in Afghanistan [32], and a study done in Ghana [33]. This could be due to the difference in the accessibility of healthcare institutions between urban and rural areas, accessing health facilities at a near distance motivates mothers to get their children vaccinated but a geographically long distance to reach health facilities demotivates mothers to get their children immunized [28].

A statistically significant association between place of delivery and complete basic childhood vaccination was observed. Children born in the health facility had higher odds of being completely vaccinated than children born at home. Supported by the study done in Tanzania [34], and a study done in east Africa [9]. The possible reason might be due to children born at health facilities getting Polio 0 and BCG vaccines at birth and mothers may get further information about full childhood immunization with their appropriate schedules [35].

Our analysis revealed region was significantly associated with complete basic childhood vaccination among children aged 12–23 months. Children residing in Afar, Somalia, and Oromia regions were less likely to be vaccinated compared to children living in Addis Ababa. It is supported by a previous study done in Ethiopia [36], Indonesia [37], and a study done in Afghanistan [24]. This might be due to the difference in healthcare infrastructures between regions, particularly pastoralist or nomadic communities living in Afar and Somalia regions are difficult to reach immunization services within the appropriate calendar due to their seasonal movement [38].

## Conclusion

The spatial distribution of complete basic childhood vaccination among children aged 12–23 months in Ethiopia was non-random (Moran's I = 0.275696). The spatial analysis identified that complete basic childhood vaccination has a significant spatial variation across the country. The most likely significant primary clusters were detected in Afder, Liben, Shabelle, and Nogobe zones of the Somalia region; Gedeo and Sidama zones of SNNPR; and Bale, and Guji zones of the Oromia region. The second significant clusters were observed in zone 1, zone 4, and zone 5 of the Afar region and Siti zone of the northern Somali region.

Both individual and community-level factors were significantly associated with complete basic childhood vaccination among children aged 12–23 months. Individual-level predictors such as maternal age, ANC visit, religion, and place of delivery were significantly associated with complete basic childhood vaccination. Community-level determinants such as region and place of residence were associated with complete basic childhood vaccination. Developing immunization campaigns by considering areas that had low complete basic childhood immunization coverage and designing healthcare programs that can motivate facility-based delivery, ANC follow-up, and enhancing maternal education is recommended.

## Limitation

There are some limitations to this study. Since the study was cross-sectional, it doesn't show the cause and effect relationship between basic childhood vaccination and its predictors. Variables such as media exposure and paternal education were not included in this study because data about such variables were not found in the 2019 EMDS, and maybe they are significant predictors of complete basic childhood vaccination.

## Acknowledgments

The authors would like to thank the MEASURE DHS program for providing the data for further analysis.

## Author Contributions

**Conceptualization:** Getanew Aschalew Tesfa, Delelegn Emwodew Yehualashet, Addisu Getnet, Kirubel Biweta Bimer, Binyam Tariku Seboka.

**Data curation:** Getanew Aschalew Tesfa, Delelegn Emwodew Yehualashet, Kirubel Biweta Bimer, Binyam Tariku Seboka.

**Formal analysis:** Getanew Aschalew Tesfa.

**Funding acquisition:** Getanew Aschalew Tesfa, Delelegn Emwodew Yehualashet, Kirubel Biweta Bimer, Binyam Tariku Seboka.

**Investigation:** Getanew Aschalew Tesfa, Delelegn Emwodew Yehualashet, Addisu Getnet, Kirubel Biweta Bimer, Binyam Tariku Seboka.

**Methodology:** Getanew Aschalew Tesfa, Delelegn Emwodew Yehualashet.

**Project administration:** Getanew Aschalew Tesfa, Binyam Tariku Seboka.

**Resources:** Getanew Aschalew Tesfa, Delelegn Emwodew Yehualashet, Addisu Getnet, Kirubel Biweta Bimer, Binyam Tariku Seboka.

**Software:** Getanew Aschalew Tesfa, Binyam Tariku Seboka.

**Supervision:** Getanew Aschalew Tesfa, Delelegn Emwodew Yehualashet, Addisu Getnet, Kirubel Biweta Bimer, Binyam Tariku Seboka.

**Validation:** Getanew Aschalew Tesfa, Delelegn Emwodew Yehualashet, Kirubel Biweta Bimer, Binyam Tariku Seboka.

**Visualization:** Getanew Aschalew Tesfa, Kirubel Biweta Bimer, Binyam Tariku Seboka.

**Writing – original draft:** Getanew Aschalew Tesfa.

**Writing – review & editing:** Getanew Aschalew Tesfa, Delelegn Emwodew Yehualashet, Addisu Getnet, Kirubel Biweta Bimer, Binyam Tariku Seboka.

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
