## [Decision Letter · Decision Letter 0]

1 May 2022

PONE-D-21-18735Spatial distribution of complete basic childhood vaccination and associated factors among children aged 12-23 months in Ethiopia. a spatial and multilevel analysis.PLOS ONE

Dear Dr. Tesfa,

Thank you for submitting your manuscript to PLOS ONE. After careful consideration, we feel that it has merit but does not fully meet PLOS ONE’s publication criteria as it currently stands. Therefore, we invite you to submit a revised version of the manuscript that addresses the points raised during the review process.

Minor revision as per the comments of the reviewers is required.  Please submit your revised manuscript by Jun 15 2022 11:59PM. If you will need more time than this to complete your revisions, please reply to this message or contact the journal office at plosone@plos.org. Please include the following items when submitting your revised manuscript:A rebuttal letter that responds to each point raised by the academic editor and reviewer(s). You should upload this letter as a separate file labeled 'Response to Reviewers'.A marked-up copy of your manuscript that highlights changes made to the original version. You should upload this as a separate file labeled 'Revised Manuscript with Track Changes'.An unmarked version of your revised paper without tracked changes. You should upload this as a separate file labeled 'Manuscript'.If applicable, we recommend that you deposit your laboratory protocols in protocols.io to enhance the reproducibility of your results. Protocols.io assigns your protocol its own identifier (DOI) so that it can be cited independently in the future. For instructions see: https://journals.plos.org/plosone/s/submission-guidelines#loc-laboratory-protocols. Additionally, PLOS ONE offers an option for publishing peer-reviewed Lab Protocol articles, which describe protocols hosted on protocols.io. Read more information on sharing protocols at https://plos.org/protocols?utm_medium=editorial-email&utm_source=authorletters&utm_campaign=protocols.

We look forward to receiving your revised manuscript.

Kind regards,

Ejaz Ahmad Khan, M.D, MPH, FFPH

Academic Editor

PLOS ONE

Journal Requirements:

2. Please provide additional details regarding participant consent. In the Methods section, please ensure that you have specified (1) whether consent was informed and (2) what type you obtained (for instance, written or verbal). If your study included minors, state whether you obtained consent from parents or guardians. If the need for consent was waived by the ethics committee, please include this information.

3. We note that Figure 2 in your submission contain map images which may be copyrighted. All PLOS content is published under the Creative Commons Attribution License (CC BY 4.0), which means that the manuscript, images, and Supporting Information files will be freely available online, and any third party is permitted to access, download, copy, distribute, and use these materials in any way, even commercially, with proper attribution. For these reasons, we cannot publish previously copyrighted maps or satellite images created using proprietary data, such as Google software (Google Maps, Street View, and Earth). For more information, see our copyright guidelines: http://journals.plos.org/plosone/s/licenses-and-copyright.

Reviewers' comments:

Reviewer's Responses to Questions

**Comments to the Author**

1. Is the manuscript technically sound, and do the data support the conclusions?

Reviewer #1: Yes

Reviewer #2: Yes

2. Has the statistical analysis been performed appropriately and rigorously? 

Reviewer #1: Yes

Reviewer #2: No

3. Have the authors made all data underlying the findings in their manuscript fully available?

Reviewer #1: Yes

Reviewer #2: Yes

4. Is the manuscript presented in an intelligible fashion and written in standard English?

Reviewer #1: Yes

Reviewer #2: Yes

5. Review Comments to the Author

Reviewer #1: The manuscript reports the analysis of data regarding vaccination coverage in Ethiopia. The text is clear and the analysis well performed wuth adequate methods. The main finding is the low complete basic childhood vaccination rate with large geographical variation within the country. A further analysis allowed to identify determinants of complete vaccination, such as mothers age and primary education, religionhealth facility delivery, residence in rural area, and region of residence. I suggest to edit the text and to slightly modify the abstract, specifying that some of those factors are associated with complete vaccination (i.e. mother age or religion) while other with incomplete vaccination (i.e. some specific regions).

Reviewer #2: Manuscript title: Spatial distribution of complete basic childhood vaccination and associated factors among children aged 12-23 months in Ethiopia. a spatial and multilevel analysis.

Reviewer comments:

The authors presented an intriguing public health issue in Ethiopia. I thoroughly read the manuscript and well written. However, I have some questions for the authors that I would like to see included in the revised manuscript.

1. The manuscript contains numerous typographical errors that should be addressed in the revised manuscript.

2. You conducted a four-model multilevel binary logistic regression analysis. My main concerns are:

• Have you checked the interaction between individual level factors with the level factors? For instance, interaction between religion and regions of the participants?

• If no why? and If so, what were your findings?

3. You've used a null model (a model without any variables). What was the percentage of complete basic childhood vaccination that could be explained by level 2 factors? To what extent does one's place of residence and/or region explain complete basic childhood vaccination? What exactly was the ICC? Was it significant?

4. The manner in which you write the relationship between the individual level factor age and complete vaccination is not as simple. Please keep it short and simple for the layperson to understand.

5. How can inequalities in immunization service be used to justify low coverage of complete basic childhood vaccination? Do you mean that Ethiopia's immunization coverage is low? If this is the case, could you please provide evidence and explain how immunization coverage affects complete basic childhood immunization?

6. PLOS authors have the option to publish the peer review history of their article (what does this mean?). If published, this will include your full peer review and any attached files.

Reviewer #1: **Yes: **Giovanni Rezza

Reviewer #2: No

---

## [Author Response · Author response to Decision Letter 0]

20 May 2022

…. point by point response...….

Academic editor comments: 

1. Please ensure that your manuscript meets PLOS ONE's style requirements, including those for file naming. Thank you for the comment. We tried to revise the manuscript based on PLOS ONE's style requirements.

2. Please provide additional details regarding participant consent in the Methods section. Thank you for the comment. We explained it. Refer page 7 line 171-175.

3. Please note that PLOS ONE is unable to publish previously copyrighted maps or satellite images, or images created using proprietary data. For these reasons, we cannot publish images generated by software which copyrights their output (such as Google Maps, Street View, and Earth). In order to use these images in your submission, we require explicit permission from the copyright owner to publish the figures under the CC BY 4.0 license.

At this time, please kindly clarify the following regarding Figure 2:

a) Where did the authors obtain the maps, basemaps, shapefiles, map data, etc. in Figure 2?

b) If any of the map images have been previously copyrighted, we require specific consent from the copyright holder to publish these images in PLOS ONE, under the CC BY 4.0 license.

Authors' response: Editor, thank you for the concern. Figure 2 is not copyrighted rather we have done using ArcGIS and SaTScan software based on the shapefile of Ethiopia received from Ethiopian Central Statistical Agency (CSA) and GPS data (longitude and latitude) from the measure demographic and health survey (DHS) program by explaining the objective of the study through online requesting and allowed us to access the shapefile and GPS data. Because we need to analyze the spatial distribution of complete basic childhood vaccination, we cited the shapefile's source. Therefore, the map presented in our study is not copyrighted; rather it shows the result of our spatial analysis.

c) Were any sample data used in the map in Figure 2 proprietary data (e,g,. from ArcGIS and/or SaTScan)?

Authors’ response: Thank you for the concern, editor. We used ArcGIS and SaTScan statistical software to illustrate the spatial distribution of complete basic childhood vaccination using data from the measure DHS program. After describing the objective of using the data, we obtained the data and acquired approval from the Measure DHS program. The DHS dataset is open to the public, and figures obtained as a final analysis result are not copyrighted.

4. Please review your reference list to ensure that it is complete and correct. Thank you. We tried to review it carefully.

Reviewer #1:

1. I suggest to edit the text and to slightly modify the abstract, specifying that some of those factors are associated with complete vaccination (i.e. mother age or religion) while other with incomplete vaccination (i.e. some specific regions). Thank you very much for the comments. We tried to revise it carefully. 

Reviewer #2:

1. The manuscript contains numerous typographical errors that should be addressed in the revised manuscript. Thank you very much for the comment. We tried to make a substantial revision on the typography.

2. You conducted a four-model multilevel binary logistic regression analysis. My main concerns are:

• Have you checked the interaction between individual level factors with the level factors? For instance, interaction between religion and regions of the participants?

• If no why? and If so, what were your findings? Thank you for the comment. Refer page 6 line 156-159.

3. You've used a null model (a model without any variables). What was the percentage of complete basic childhood vaccination that could be explained by level 2 factors? To what extent does one's place of residence and/or region explain complete basic childhood vaccination? What exactly was the ICC? Was it significant? Thank you for your interesting comment. We included it based on your comment. Refer page 13 line 253-259 and Table 4 page 16.

4. The manner in which you write the relationship between the individual level factor age and complete vaccination is not as simple. Please keep it short and simple for the layperson to understand. Thank you for the comment. We rephrased it. Refer page 13 line 270-273.

5. How can inequalities in immunization service be used to justify low coverage of complete basic childhood vaccination? Do you mean that Ethiopia's immunization coverage is low? If this is the case, could you please provide evidence and explain how immunization coverage affects complete basic childhood immunization? Thank you for the comment. We corrected it. Refer page 17 line 306-311.

---

## [Editor Report · Decision Letter 1]

25 Oct 2022

PONE-D-21-18735R1Spatial distribution of complete basic childhood vaccination and associated factors among children aged 12-23 months in Ethiopia. a spatial and multilevel analysis.PLOS ONE

Dear Dr. Tesfa,

Thank you for submitting your manuscript to PLOS ONE. After careful consideration, we feel that it has merit but does not fully meet PLOS ONE’s publication criteria as it currently stands. Therefore, we invite you to submit a revised version of the manuscript that addresses the points raised during the review process.

We look forward to receiving your revised manuscript.

Kind regards,

Demisu Zenbaba Heyi, MPH

Academic Editor

PLOS ONE

Journal Requirements:

Additional Editor Comments:

Please, address previous reviewer comments properly e.g. why you not calculated the PCV(proportion of change in variance for Model I, II, II), and better to calculate the median odds ratio for all models as well?

---

## [Author Response · Author response to Decision Letter 1]

7 Nov 2022

General comment (Journal Requirements)

1. Please review your reference list to ensure that it is complete and correct. If you have cited papers that have been retracted, please include the rationale for doing so in the manuscript text, or remove these references and replace them with relevant current references.

Response: thank you very much for your suggestion. We tried to revise our reference and change was made on the following references.

- L Arevshatiana, C.C., SK Lwangac, AO Misored, P Ndumbee, JF Sewardf, P Taylorg. An evaluation of infant immunization in Africa: is a transformation in progress? ; Available from: https://www.who.int/bulletin/volumes/85/6/06-031526/en/.: Removed and Replaced by: Lindstrand, A., et al., The World of Immunization: Achievements, Challenges, and Strategic Vision for the Next Decade. The Journal of Infectious Diseases, 2021. 224(Supplement_4): p. S452-S467.

- Organization, W.H. Immunization, Vaccines and Biologicals. Available from: https://www.who.int/immunization/documents/immunological_basis_series/en/. : removed

- Ministry of Health-Ethiopia. Expanded Program on Immunization (EPI). Available from: https://www.moh.gov.et/ejcc/am/EPI. Removed and replaced by: Belete, H., et al., Routine immunization in Ethiopia. The Ethiopian Journal of Health Development, 2015. 29.

- 14. Federal Ministry of Health, E. ETHIOPIA NATIONAL EXPANDED PROGRAMME ON IMMUNIZATION COMPREHENSIVE MULTI-YEAR PLAN 2011 - 2015. 2010; Available from: https://bidinitiative.org/wpcontent/files_mf/1405630243EthiopiaComprehensivemultiyearplanfor20112015Year2010.pdf. Removed and replaced by: Boulton, M.L., et al., Vaccination timeliness among newborns and infants in Ethiopia. PLOS ONE, 2019. 14(2): p. e0212408.

Editor comment

2. Please, address previous reviewer comments properly. Why you not calculated the PCV (proportion of change in variance for Model I, II, II), and better to calculate the median odds ratio for all models as well? 

Response: Thank you for your insight. We have already addressed it. Please refer page 14, line 280-290 and Table 4, page 16.

---

## [Editor Report · Decision Letter 2]

14 Nov 2022

PONE-D-21-18735R2Spatial distribution of complete basic childhood vaccination and associated factors among children aged 12-23 months in Ethiopia. a spatial and multilevel analysis.PLOS ONE Dear Dr. Tesfa,

Thank you for submitting your manuscript to PLOS ONE. After careful consideration, we feel that it has merit but does not fully meet PLOS ONE’s publication criteria as it currently stands. Therefore, we invite you to submit a revised version of the manuscript that addresses the points raised during the review process.

The following previous reviewer comments are not addressed:Have you checked the interaction between individual level factors with the level factors? For instance, interaction between religion and regions of the participants?If no why? and If so, what were your findings?The manner in which you write the relationship between the individual level factor age and complete vaccination is not as simple. Please keep it short and simple for the layperson to understand.How can inequalities in immunization service be used to justify low coverage of complete basic childhood vaccination? Do you mean that Ethiopia's immunization coverage is low? If this is the case, could you please provide evidence and explain how immunization coverage affects complete basic childhood immunization?

We look forward to receiving your revised manuscript.

Kind regards,

Demisu Zenbaba Heyi, MPH

Academic Editor

PLOS ONE

---

## [Author Response · Author response to Decision Letter 2]

17 Nov 2022

We note that your Data Availability statement states the following: "Data is available online from https://www.dhsprogram.com/data. A letter of permission for the use of the data was secured from DHS program. We used 2019 EMDHS dataset."

Before we can proceed, please clarify if the authors had special access to the data that other researchers would not have.

If not, please also provide a direct link to the dataset.

Response: The data is publicly available and anyone can access it through reasonable request. The sole requirements for accessing the Demographic and Health Survey (DHS) program datasets are registration and a justification of the request's objective. 

….point by point response...

1. Have you checked the interaction between individual level factors with the level factors? For instance, interaction between religion and regions of the participants? If no why? and If so, what were your findings?

Response: Thank you for your comment. Understanding geographical distributions of complete basic childhood vaccination and identifying associated predictors is crucial for designing effective policies and strategies that help to improve vaccination coverage. Keep in mind this, the first main specific objective of our study was to identify the spatial distribution of complete basic childhood vaccination. To attain this objective and to answer the question of where the hotspots for low vaccinations are located, spatial statistical analysis is required. So, our spatial analysis looked at only the dependent variable. Second, to identify potential predictor variables, regression analysis is required. Hence, we used multilevel multivariable logistic regression analysis. As stated above we identified predictors using regression analysis and regression models consider linear relationships (stationary relationships). Since one of the assumptions of regression models is that relationships are constant across the entire study area. Therefore, estimates generated in this model assume averages. Due to that, we didn't consider the interaction between individual-level factors with the level factors. However, we have checked the absence of multicollinearity between these independent variables using the variance inflation factor.

2. The manner in which you write the relationship between the individual level factor age and complete vaccination is not as simple. Please keep it short and simple for the layperson to understand.

Response: we revised it accordingly and corrected it as follows

“Maternal age was significantly associated with complete basic childhood vaccination. Children born to mothers who were 20-24 years old were approximately two times (AOR=2.12, 9S%CI: 1.06-4.24) more likely to be fully vaccinated than children whose mothers were 15-19 years old.” Refer page 13, line 263-266 (on the tracked version)

3. How can inequalities in immunization service be used to justify low coverage of complete basic childhood vaccination? Do you mean that Ethiopia's immunization coverage is low? If this is the case, could you please provide evidence and explain how immunization coverage affects complete basic childhood immunization?

Response: Thank you for your constructive comment. We mean that variability of immunization service access across areas might be one of the reasons for the low prevalence of complete basic childhood vaccination. Since our study was done based on the national representative data that took into account both rural and urban areas, the overall prevalence of complete basic childhood vaccination was varied compared to previous studies that were done in specific districts. Anyways, in order to minimize such type of misunderstanding, we have made some modifications in that sentence and additional explanations were added in the main document. Refer page 17, line 305-309 (on the tracked version)

---

## [Editor Report · Decision Letter 3]

7 Dec 2022

Spatial distribution of complete basic childhood vaccination and associated factors among children aged 12-23 months in Ethiopia. a spatial and multilevel analysis.

PONE-D-21-18735R3

Dear Dr. Tesfa,

We’re pleased to inform you that your manuscript has been judged scientifically suitable for publication and will be formally accepted for publication once it meets all outstanding technical requirements.

Kind regards,

Demisu Zenbaba Heyi, MPH

Academic Editor

PLOS ONE
---

## [Editor Report · Acceptance letter]

27 Dec 2022

PONE-D-21-18735R3 

Spatial distribution of complete basic childhood vaccination and associated factors among children aged 12-23 months in Ethiopia. A spatial and multilevel analysis. 

Dear Dr. Tesfa:

I'm pleased to inform you that your manuscript has been deemed suitable for publication in PLOS ONE. Congratulations! Your manuscript is now with our production department. 

Kind regards, 

on behalf of

Dr. Demisu Zenbaba Heyi 

Academic Editor

PLOS ONE